

# On the Estimation of Global Plant Water Requirement

Yunfei Wang[1,2], Yijian Zeng[2], Zengjing Song[2], Danyang Yu[2], Qianqian Han[2], Enting Tang[2], Henk de Bruin[3], and Zhongbo (Bob) Su[2]

[1]School of Water Conservancy and Transportation, Zhengzhou University, Zhengzhou, 450001, China
5  [2]Faculty of Geo-Information Science and Earth Observation, University of Twente, Enschede, 7522 NB, The Netherlands
[3]Meteorology and Air Quality, Wageningen University, Wageningen, 6708 PB, The Netherlands

*Correspondence to*: Zhongbo (Bob) Su and (z.su@utwente.nl) and Yunfei Wang (y.wang-3@utwente.nl)

**Abstract.** Water supply is the most critical constraint for vegetation growth and food security. The amount of water demand by plant growth is usually estimated by plant water requirement which unfortunately cannot be directly measured at any 10  large scale in field conditions. Different estimation methods have been proposed in the past seven decades for estimating plant water requirements using the concept of reference evapotranspiration (ET0) methods or potential evapotranspiration (PET) methods. In addition, using PET or ET0 to estimate actual evapotranspiration (ETa) is a critical approach in hydrological and climate models. However, different PET or ET0 models provide diverse results for irrigation water requirement (IWR) that in turn may result in a huge waste of irrigation water. Here, we assess the suitability of six common 15  methods for estimating PET at 170 eddy covariance flux sites and propose a practical approach for estimating the IWR using a physically consistent model STEMMUS-SCOPE.

Notably, the Priestley-Taylor and LSA_SAF method excels in providing reasonable approximations of daily PET. Consequently, in scenarios where net radiation data and ground heat flux are accessible, the Priestley-Taypor method emerges as the recommended choice. The LSA_SAF method is the better one when only net radiation data is available. 20  Alternatively, in cases where only global radiation data is available, the Makkink and Hargreaves methods serve as viable substitutes. Although the FAO56 Penman-Monteith method is much better than the original Penman-Monteith method when wind speed and air humidity data are at hand, its suitability falls short of the preferred status. This study contributes to understanding and quantifying the applicability of different methods in estimating PET and IWR, based on input data availability and physical considerations.

## 1 Introduction

Evapotranspiration (ET) is a complex process that proves challenging to measure directly (Jensen and Allen, 2016; Vremec et al., 2023a; Vremec et al., 2023b; Wang and Dickinson, 2012). Consequently, it is commonly estimated by (semi-)empirical formulas based on more readily available meteorological observations, including air temperature and humidity, wind speed, and radiation (global radiation or net radiation).



For decades, numerous methods have been proposed and implemented to estimate ET. These methods typically yield varying estimates of evapotranspiration due to differences in methodologies and data sources (Lemaitre-Basset et al., 2022; McMahon et al., 2013). Many of these formulas are designed to estimate either potential evapotranspiration, which signifies the maximum ET under optimal water availability conditions (Xiang et al., 2020), or reference crop evapotranspiration, representing ET from a reference surface or crop with ample water supply (Allen et al., 1998).

Potential evapotranspiration (PET) is commonly defined as the quantity of water that has the potential to evaporate and transpire from a vegetated landscape, solely constrained by atmospheric demand and without any additional limitations (Guan et al., 2021; Singer et al., 2021; Jensen and Allen, 2016). Accurate estimates of PET play a crucial role in numerous applications, including hydrological, ecological, and land surface models extensively used in global change research (Lu et al., 2005). Discrepancies in PET calculations have a cascading effect throughout a modeling sequence, ultimately affecting

the outcomes of a study (Lemaitre-Basset et al., 2022). Many studies demonstrated that climate change impact assessments are affected by the choice of PET methods (Bormann, 2010; Prudhomme and Williamson, 2013). Similarly, highly efficient agricultural water management hinges on PET calculations, rendering it susceptible to the employed methodologies (Kumar et al., 2012).

Similarly, the concept of the reference evapotranspiration (ET0) holds pivotal significance within water balance calculations

and surface energy assessments. Accurate knowledge of ET0 proves indispensable for irrigation engineers, cultivators, water resource administrators, and policymakers who engage with irrigation design, water distribution systems, and resource management. Conventionally, lysimeters constituted the primary means for approximating ET0. However, contemporary circumstances curtail the deployment of lysimeters for ET0 estimation, attributed to their elevated installation, maintenance, and operational expenditures. In response, diverse mathematical models have emerged as more pragmatic avenues for

indirectly estimating ET0, rendering them preferred substitutes to direct methodologies due to their user-friendliness. Across various research endeavors, semi-empirical methods and process-based models have demonstrated noteworthy prolificacy in ET0 estimation by leveraging the limited climatic variables as inputs.

Numerous studies have employed also the ET0 estimate by the FAO-56 Penman-Monteith (FAOPM) model to assess the dependability of PET models, a practice that warrants correction (Li et al., 2016). Many investigations have resorted to the

FAOPM model as a benchmark for evaluating alternative PET methods due to the absence of measured actual evapotranspiration (ETa) data. However, it is worth noting that the FAOPM method is an empirical estimation technique, not a direct measurement approach for ET0. McMahon et al. (2013) have shed light on this matter, clarifying that the FAOPM method should not be considered a reference.

Nevertheless, the FAOPM method has been extensively employed for global ET0 quantification (Bjarke et al., 2023; Yan et

al., 2023; Zomer et al., 2022) but several studies have highlighted also that the FAOPM method yields unrealistic ET0 estimations in desert regions, such as the Sahara (Aschonitis et al., 2022; Wang and Dickinson, 2012). The FAOPM formula prescribes a constant surface resistance (70 s m-1), neglecting its interaction with air humidity. In essence, the FAOPM method overlooks the intricate relationship between vegetation and the atmosphere, while substantial research has



demonstrated that stomatal resistance is regulated by both soil moisture and vapor pressure deficit (VPD) (Wang et al., 2022;

Zhang et al., 2021). Even though the FAOPM formulation assumes well-watered reference vegetation (grass or alfalfa), VPD significantly influences stomatal closure, particularly when applied across diverse climatic conditions. Consequently, the crop coefficient necessitates adjustment according to varying locales and climates (Pereira et al., 2021), thus diminishing the method's intended universality. This discrepancy constitutes the primary reason behind the reported unrealistically high ET0 values by the FAOPM method (Alexandris and Proutsos, 2020; Aschonitis et al., 2022).

In this research, we employed the advanced STEMMUS-SCOPE model, known for its process-based and physical consistency, to replicate the actual evapotranspiration (ETas) and water stress factor (WSF). Following this, the potential evapotranspiration (ET0s) was computed by the ETas to WSF ratio constrained by available energy. The ET0 values derived from the STEMMUS-SCOPE model were utilized to evaluate the six conventional ET0 or PET methods at 170 flux sites worldwide. Ultimately, the irrigation water requirements (IWR for crops, or the insatiate water requirement of plant growth)

were calculated by the ET0s and ETas simulated by STEMMUS-SCOPE. The objective of this investigation is to assess the validity of different PET methods and offer a practical approach for calculating IWR.

## 2 Datasets and Methods

### 2.1 Description of PLUMBER2 dataset

The meteorological forcing of this study is from the PLUMBER2 dataset. PLUMBER2 is the second phase of the "Protocol

for the Analysis of Land Surface Models (PALS) Land Surface Model Benchmarking Evaluation Project". PLUMBER2 conducted a multi-model (more than 20 land surface or biosphere models) intercomparison (Ukkola et al., 2022). For driving land surface models, fully gap-filled meteorological data of the 170 sites are provided after quality control. Additional meta-data, such as site descriptions, reference and canopy heights, plant functional types, and satellite leaf area index are also provided. The distribution of the 170 flux sites is shown in Fig. S1. The detailed information of these sites is shown in Table

S2.

### 2.2 The simulation of ETa and WSF with STEMMUS-SCOPE

The STEMMUS-SCOPE model integrates a comprehensive canopy radiative transfer, energy balance, and photosynthesis model (SCOPE) (Van der Tol et al., 2009) with a two-phase vadose zone mass and heat transfer model (STEMMUS) (Yu et al., 2018; Zeng and Su, 2013; Zeng et al., 2011a; Zeng et al., 2011b). It considers carbon assimilation by the canopy and

subsequent carbon allocation to shoots and roots to synchronize the exchanges of water, energy, and $CO_2$ between the atmosphere and the soil, thereby modulating the dynamics of soil water and heat transport. Therefore, STEMMUS-SCOPE can simulate the transfer of optical, thermal, and fluorescent radiation, water flux, and carbon fluxes within the Soil-Plant-Atmosphere Continuum (SPAC) system, and it can be used to produce the eco-hydrological dataset across different vegetation types (Wang et al., 2021).





## 2.3 Different methods for calculating reference and/or potential evapotranspiration (ET0)

Six common PET methods (two net radiation-based models, two resistance-based models, and two global radiation & temperature-based models) are used in this study, including MAK, HRG, FAOPM, PM, PT, and LSA_SAF (Table 1). The HRG model is a modified version of MAK model after considering advection (Cruz-Blanco et al., 2014). FAOPM is developed by the Food and Agriculture Organization of the United Nations (FAO) as part of its "Crop Evapotranspiration" guidelines, commonly known as FAO-56, which provide standardized methods for estimating evapotranspiration for agricultural and environmental purposes.

### 2.3.1 The Makkink formula (MAK) (Bruin, 1987)

$\lambda ET0_{MAK}$ is calculated following Eq. (1):

$$\lambda ET0_{MAK} = 0.65 \frac{\Delta}{\Delta+\gamma} R_s \ , \tag{1}$$

where $\Delta$ is the slope of the saturation water vapor curve, it is a function of air temperature; $\gamma$ is the psychrometer constant (see FAO56); Rs is the daily mean global radiation.

### 2.3.2 The Hargreaves (HRG) formula (Hargreaves and Samani, 1985)

$\lambda ET0_{HRG}$ is calculated following Eq. (1):

$$\lambda ET0_{HRG} = 0.0135 R_s (T_a + 17.8) \ , \tag{2}$$

where $T_a$ is the air temperature; $R_s$ the daily mean global radiation.

### 2.3.3 The revised Penman-Monteith formula in FAO56 report (PMFAO) (Allen et al., 1998)

$\lambda ET0_{PMFAO}$ is calculated following Eq. (1):

$$\lambda ET0_{PMFAO} = \frac{0.408\Delta(R_n - G)0.0864 + \gamma\frac{900}{T_a+273}U_2(e_s - e_a)}{\Delta + \gamma(1 + 0.34 U_2)} \ , \tag{3}$$

where $R_n$ is net radiation (W m$^{-2}$); $G$ is the ground heat flux (W m$^{-2}$); $\Delta$ is the slope of the saturation water vapor curve; $\gamma$ is the psychrometer constant (kPa °C$^{-1}$); $e_a$ is the actual vapor pressure (kPa); $e_s$ is the saturated vapor pressure (kPa); $U_2$ is the wind speed at 2 m height (m s$^{-1}$).

### 2.3.4 The Penman-Monteith formula (PM) (Monteith, 1965)

$\lambda ET0_{PM}$ is calculated following Eq. (1):

$$\lambda ET0_{PM} = \frac{\Delta(R_n - G) + \rho c_p(e_s - e_a)g_a}{\left(\Delta + \gamma\left(1 + \frac{g_a}{g_s}\right)\right)} \ , \tag{4}$$



where the definitions of $R_n$, $G$, $\Delta$, $\gamma$ are the same as those mentioned above. In addition, $\rho$ is air density (1.2 kg m$^{-3}$); $c_p$ is the specific heat of the air (1004.7 J kg$^{-1}$ K$^{-1}$); $g_s$ is surface conductance (m s$^{-1}$); $g_a$ is aerodynamic conductance (m s$^{-1}$). It is challenging to determine $g_s$, however if we assume that the surface is not short of water (i.e. a wet surface), $g_s$ can be set as infinity, i.e. the surface resistance to ET is zero. For comparison purpose, we choose the surface resistance as $r_s = 70$ s m$^{-1}$ ($g_s = 1/r_s$) following the FAO recommendation for a short grass.

**2.3.5 The Priestley-Taylor formula (PT) (Priestley and Taylor, 1972)**

$\lambda ET0_{PT}$ is calculated following Eq. (1):

$$\lambda ET0_{PT} = 1.26(R_n - G)\frac{\Delta}{\Delta+\gamma} \,, \tag{5}$$

the definitions of $R_n$, $G$, $\Delta$, $\gamma$ are the same as those mentioned above.

**2.3.6 The LSA_SAF formula (De Bruin et al., 2016; Trigo et al., 2018)**

$\lambda ET0_{HRG}$ is calculated following Eq. (1):

$$\lambda ET0_{LSA\,SAF} = R_n\frac{\Delta}{\Delta+\gamma} + \beta \,, \tag{6}$$

where the definitions of $R_n$, $G$, $\Delta$, $\gamma$ are the same as those mentioned above. $\beta$ is a constant (20 Wm$^{-2}$) introduced to compensate for deviations in near-surface conditions from fully saturated air.

**2.4 Calculation of potential ET with STEMMUS-SCOPE**

Because of the difficulty or rather impossibility of measuring the potential ET in field conditions, we utilize the simulated ETa and water stress factor (WSF) by STEMMUS-SCOPE to derive PET. We assume implicitly that PET equals to the ETa when available water for the plant is not a limiting factor, thus PET is only limited by available energy. With the WSF and ETa simulated by STEMMUS-SCOPE, we can calculate the PET with the equation as follows:

$$ET0_S = \frac{ETas}{WSF} \,, \tag{7}$$

$$if\ \lambda ET0_S > R_a : \lambda ET0_S = R_a \,, \tag{8}$$

where $ET0_S$ is the potential ET calculated by STEMMUS-SCOPE, $ET_{as}$ is the actual ET simulated by STEMMUS-SCOPE which is constrained by available energy ($Ra$), $WSF$ is the water stress factor simulated by STEMMUS-SCOPE model (its calculation can refer to Wang et al. 2021, eq. A11 and A12). $ET_{as}$ and $WSF$ can be evaluated at the 170 flux sites.



## 2.5 Calculation of Irrigation Water requirement (IWR)

We aim to assess the suitability and easy applicability of six methods for estimating ET0. We follow the definition of the Food and Agricultural Organization (FAO) of the United Nations for the various concepts for estimating IWR.

The net IWR refers to the amount of water that needs to be supplied to agricultural crops or plants through artificial means (irrigation) to meet their water needs for optimal growth and production. It takes into account factors such as climate, soil type, crop type, evapotranspiration rates, and local conditions. The calculation of IWR typically involves estimating the

difference between the water lost through evapotranspiration by the plants and the effective rainfall received in the area. The formula generally follows the following equations (all values are depths, in mm):

$$P + IWR = ET0 - PR + (M1 - M0) + R , \tag{9}$$

$$P = ETa - PR + (M1 - M0) + R , \tag{10}$$

$$IWR = ET0 - ETa , \tag{11}$$

Where, $IWR$ - Irrigation water requirement (mm); $P$ - Precipitation (mm); $ET0$ - Potential evapotranspiration estimated by STEMMUS-SCOPE and constrained by available energy (mm); $ETa$ - Actual evapotranspiration estimated by STEMMUS-SCOPE and constrained by available energy (mm); $PR$ - Percolation and recharge to groundwater (mm); $ΔM$ - Soil moisture change, $ΔM = M1 - M0$, from time 0 to time 1 in the root zone; $R$ - Runoff (mm).

**Table 1. Variables and parameters required by different PET formulas.**

| Method | Temperature | Radiation | Humidity | Wind speed | Others |
|---|---|---|---|---|---|
| MAK (Bruin, 1987) | Ta | Rs | | | |
| HRG (Hargreaves and Samani, 1985) | Ta | Rs | | | |
| PMFAO (Allen et al., 1998) | Ta | Rn, G | VPD | $U_2$ | |
| PM (Monteith, 1965) | Ta | Rn, G | VPD | $U_z$ | |
| PT (Priestley and Taylor, 1972) | Ta | Rn, G | | | Calibration constant (1.26) |
| LSA_SAF (Trigo, I.F. et al., 2018) | Ta | Rn | | | Calibration constant $\beta$ |

## 2.6 Statistical analysis

The statistics we used to evaluate the performance of the model were (1) Coefficient of determination ($R^2$); (2) Mean bias error (MBE); (3) Standard deviation (SD); and Normalized mean error (NME). They can be calculated as follows:




$$R^2 = 1 - \frac{\sum_{i=1}^{n}(O_i - M_i)^2}{\sum_{i=1}^{n}(O_i - \bar{O})^2} , \tag{11}$$

$$MBE = \frac{\sum_{i=1}^{n}(M_i - O_i)}{n} , \tag{11}$$

$$SD = \left| 1 - \frac{\sqrt{\frac{\sum_{i=1}^{n}(M_i - \bar{M})^2}{n-1}}}{\sqrt{\frac{\sum_{i=1}^{n}(O_i - \bar{O})^2}{n-1}}} \right| , \tag{11}$$

$$NME = \frac{\sum_{i=1}^{n}|M_i - O_i|}{\sum_{i=1}^{n}|\bar{O} - O_i|} , \tag{11}$$

where $M_i$ is the ith value of modelling, $O_i$ is the ith value of observation, $\bar{O}$ is the average of the observations, $\bar{M}$ is the average of the modelling, and n is the number of samples.

## 3 Results

In this section, we present results for 170 sites of different vegetation types and evaluate the goodness of fit between the modelled PET by six methods (ET0m) and PET calculated by STEMMUS-SCOPE (ET0s), using the coefficient of determination ($R^2$), the Mean bias error (MBE), the Standard deviation (SD), and the Normalized mean error (NME).

### 3.1 Statistical analysis of 170 flux sites

To investigate the consistency of the six common PET models, the correlations of ET0m and the ET0s are showed in Fig. 1. Overall, the PT and LSA_SAF method performed better than other methods. The FAOPM, MAK, and HRG methods performed relatively well. However, the $R^2$ of the PM method is significantly lower than that of other methods. Similar to Fig.1, Table S3 shows the median $R^2$, MBE, SD, and NME values of different methods in representing ET0. The PT performed best with the lowest median MBE value, the lowest SD value, the lowest NME value, and the highest median $R^2$ value, while the PM performed badly with the highest median MBE value, the highest median SD value, the highest median NME value and lowest median of $R^2$. The median $R^2$ value of LSA SAF is comparable with that of PT, while the median MBE, SD, and NME values are significantly higher than those of PT method. It is consistent with the results shown in Fig. 1.

### 3.2 Statistical analysis of different vegetation types

Fig. S2.1-2.4 presents the comprehensive analysis of the Coefficient of determination ($R^2$), Mean bias error (MBE), Normalized mean error (NME), and Standard deviation (SD) between the calculated potential evapotranspiration (ET0m) using different formulas and modeled potential evapotranspiration (ET0s) by STEMMUS-SCOPE for different vegetation types.



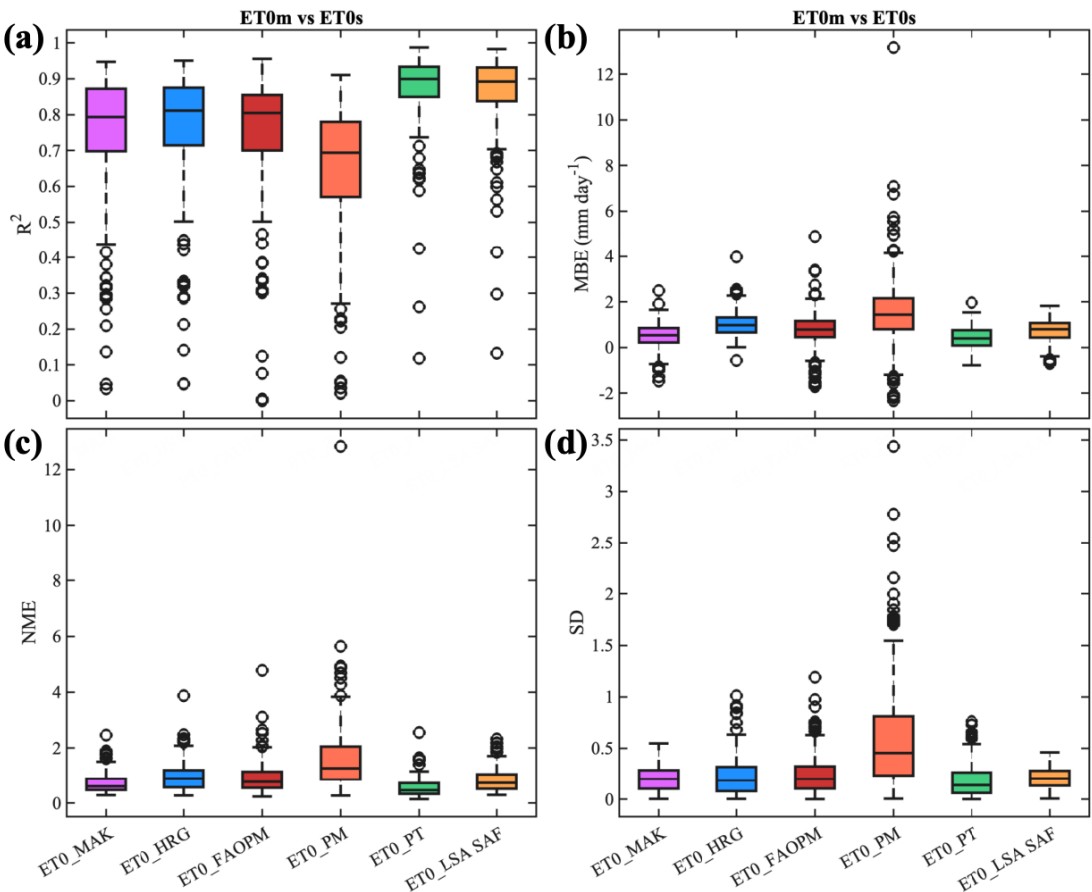

**Figure 1 The (a) Coefficient of determination ($R^2$), (b) Mean bias error (MBE, in mm day$^{-1}$), (c) Standard deviation (SD), and (d) Normalized mean error (NME) between calculated daily potential evapotranspiration (ET0$_m$) using different formulas and modeled daily potential evapotranspiration (ET0$_s$) by STEMMUS-SCOPE at 170 flux sites.**

The PT and LSA_SAF methods outperformed the other four methods across all vegetation types (including SHR: (Open/Closed) Shrublands, CRO: Croplands, DBF: Deciduous Broadleaf Forests, EBF: Evergreen Broadleaf Forests, ENF: Evergreen Needleleaf Forests, GRA: Grasslands, MF: Mix Forests, WET: Wetland, SAV: (Woody) Savannas), displaying higher median $R^2$ values, while the PM method showcased the lowest median $R^2$ values. Median MBE, NME, and SD values exhibited significant variability across different vegetation types. SHR, WET, and SAV consistently showed low MBE values, whereas DBF and ENF demonstrated notably higher values. CRO displayed lower MBE values for MAK and PT methods, while GRA and MF exhibited low values only for the PT method. MAK and FAOPM methods showed lower MBE values at EBF sites. Concerning NME, all methods displayed median NME values clustered at a low level in the SHR sites. In CRO, GRA, and MF sites, the PT method yielded the lowest median NME value. MAK, FAOPM, and LSA SAF methods consistently showed lower NME values than the PM method. Within DBF and EBF sites, NME values for MAK, FAOPM, PT, and LSA SAF methods were comparable. In ENF sites, the FAOPM method exhibited the lowest NME value, while the



PM method showed the highest. Across WET sites, all methods demonstrated low NME values, with the PT method achieving the lowest median NME, slightly outperforming other methods. For SAV sites, median NME values for MAK, HRG, and LSA SAF methods closely aligned, surpassing those associated with the PT method, while elevated NME values were observed for both FAOPM and PM methods. Regarding SD, median values for SHR fell within the range of 0.23 to 0.33. For ENF, the MAK method exhibited the lowest SD value. In SAV, CRO, and DBF, the PT method yielded the lowest median SD values, while for GRA and MF, PT showed the lowest median SD values. WET showed relatively better performance from HRG and PM methods. Lastly, DBF showed the lowest SD value for the LSA SAF method.

### 3.3 Comparison of the estimated annual PET from different methods

Fig. 2 displays the boxplot of the calculated annual PET of all stations across distinct vegetation types, estimated with various methods. As depicted in the figure, the simulated ETa by STEMMUS-SCOPE model aligns comparably with the measured ETa acquired through the eddy covariance (EC) system. Notably, for the SHR, EBF, and SAV sites, the annual PET are accurately estimated through the MAK, FAOPM, PT, and LSA_SAF methods. In the case of the CRO, MF, and GRA sites, the annual PET demonstrates comparability with the estimates derived from the PT method. For the DBF sites, the MAK and PT methods provide relatively well annual PET estimation. Concerning the ENF sites, it is noteworthy that all methods overestimated PET, even though the PT method presents the highest $R^2$ value (as shown in Fig. 4). In contrast, for the WET sites characterized by free water surfaces, all methods exhibit satisfactory agreement with the data, indicating a generally acceptable performance across the board.

### 3.4 Calculation of Irrigation Water Requirement (IWR)

The STEMMUS-SCOPE model simulates both actual evapotranspiration (ETa) and reference evapotranspiration (ET0), effectively accounting for both energy and water constraints. Consequently, it enables the calculation of irrigation water requirements (IWR, or insatiate water requirement by plant growth) using eq. (11) to (13). In this context, irrigation becomes advisable when the computed IWR surpasses zero. Fig. 3 presents the computed IWR data generated by STEMMUS-SCOPE for various vegetation types across 170 distinct sites. For SHR and SAV, the IWR consistently registers values greater than zero for all sites, signifying a significant water shortage. Conversely, for forested areas including DBF, EBF, ENF, and MF, the majority of sites exhibit no water shortage. This observation underscores that most forested sites enjoy an ample supply of natural precipitation. In the cases of CRO and GRA, the IWR exhibits considerable variation across different sites. In regions such as Australia and the United States, the majority of GRA and CRO sites show IWR values greater than zero, implying a potential need for irrigation. In contrast, across Asian and European locations, the IWR for most GRA and CRO sites tends to be zero, signifying a general sufficiency of water supply. Additionally, it is worth noting that for most sites characterized as WET, ET0 and ETa exhibit strikingly similar patterns, resulting in IWR values that equal to zero. This suggests that the STEMMUS-SCOPE model effectively approximates actual water consumption for these wetland sites. The




IWR calculations for typical cropland sites in Europe are depicted in Fig. S3.1.4-3.6.4. For example, the daily IWR of the ES-ES2 site in Spain exhibits a noticeable seasonal pattern, with IWR increasing during periods of reduced precipitation.

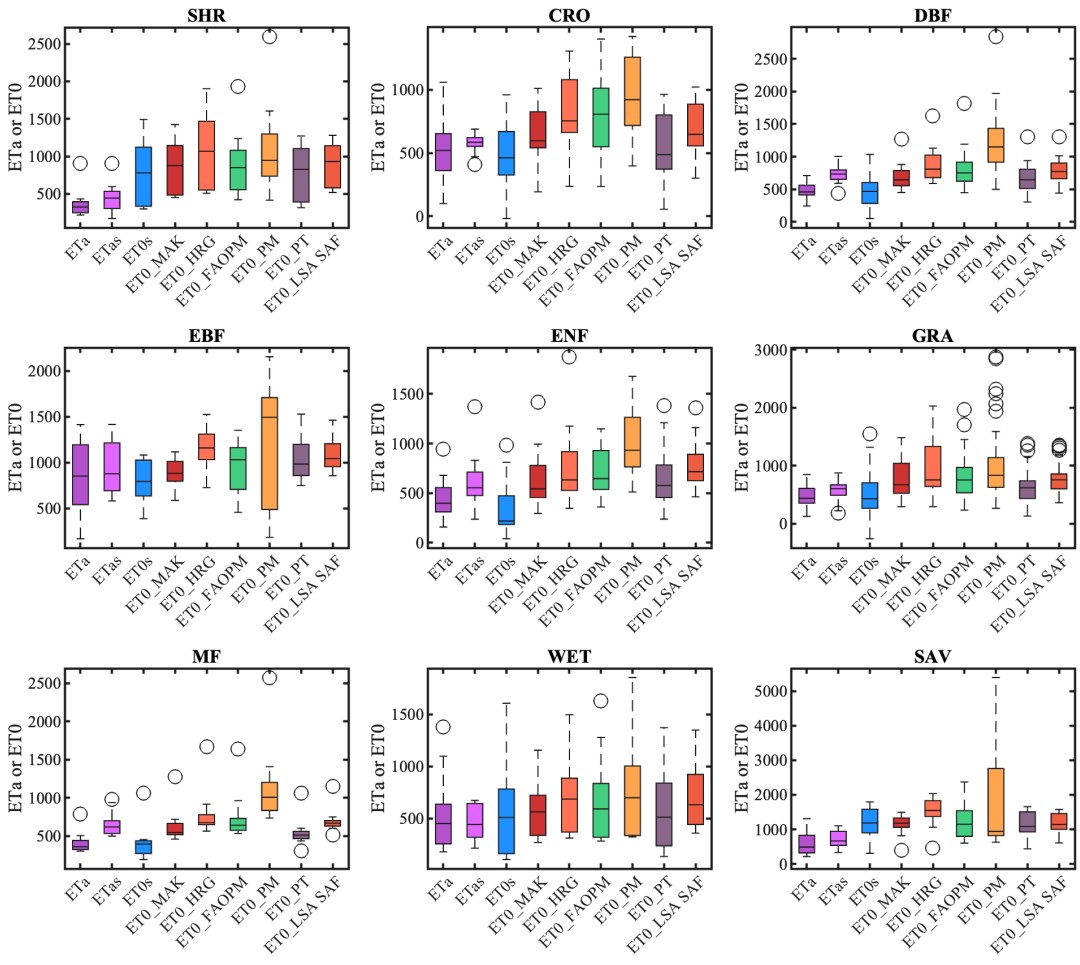

235

**Figure 2 Observed ETa, simulated ETas, calculated ET0 by STEMMUS-SCOPE (ET0s), and calculated ET0 by six methods (ET0_MAK, HRG, ET0_FAOPM, ET0_PM, ET0_PT, and ET0_HRG) for different vegetation type at 170 flux sites. (shown are annual values)**



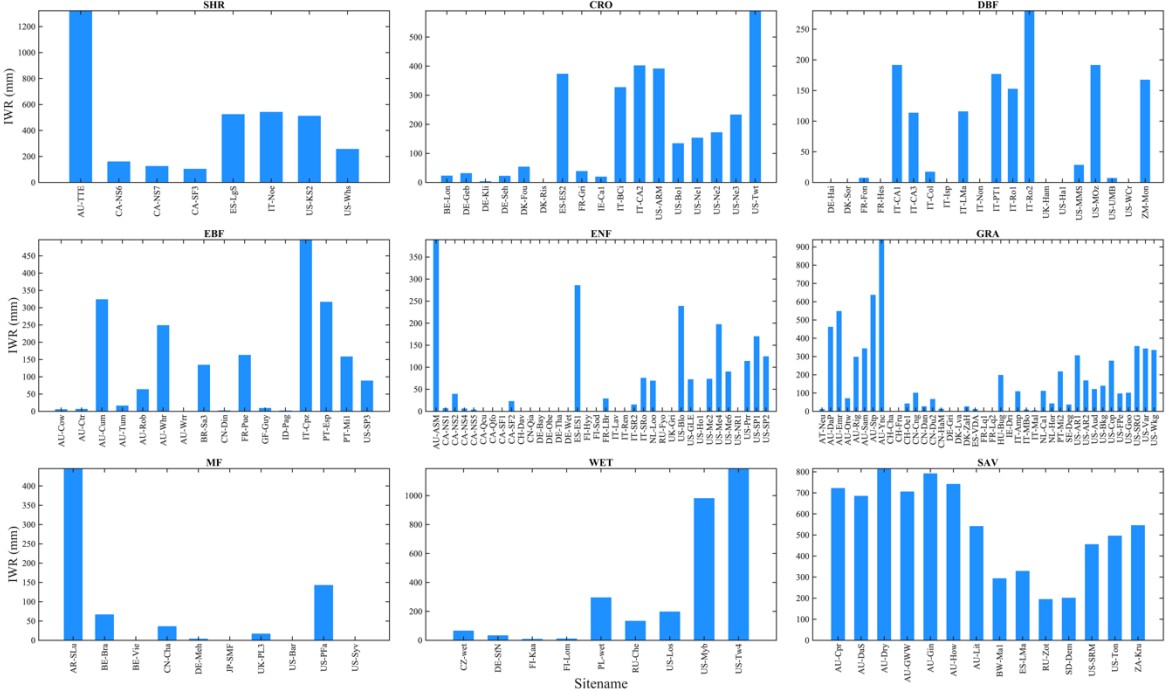

**Figure 3 The calculated irrigation water requirement (IWR) by STEMMUS-SCOPE for different vegetation types at 170 flux sites.**

## 4 Discussion

### 4.1 The reasons why the radiation-based method performed best across the world

The PT models performed best in estimating daily PET across the world (Table S3 and S4). In the absence of soil heat flux, the performance of the LSA SAF method is slightly inferior to that of the PT method. The MAK and HRG methods are satisfactory, both considering temperature in addition to global radiation. It indicates the adding of temperature has limited improvement in their application in different vegetation types. The PM methods produced large errors in most sites. Thus, this model was not suitable for estimating daily PET. Earlier research inferred that the resistance-based PET models were not suitable in the various vegetations due to the difficulty in defining the resistance (Lu et al., 2005; Tabari et al., 2013; Xu and Singh, 2002).

The PT method is commonly employed for estimating ET over open water, offering estimations that closely align with atmospheric evaporative demand (Su and Singh, 2023). Previous studies also demonstrated that the superior performance of radiation-based PET models compared to temperature-based and resistance-based models. The superiority of the PT method is attributed to the dominant influence of net radiation on land ET (Xystrakis and Matzarakis, 2011; Tabari et al., 2013; Li et al., 2016). Consequently, the PT method is considered the most reliable approach for calculating PET across different vegetation types (Fisher et al., 2009).





Lastly, this study indicates the challenging accuracy of estimating PET and advises caution in its use for estimating actual evapotranspiration. The PET methods commonly employed in this comparative analysis yielded a broad range of values. The difference was particularly pronounced at the SD-Dem site in Sudan (a savanna site, refer to Table S2), where the annual PET fluctuated as much as 4621 mm/yr. Specifically, the PT method estimated an annual PET of 770 mm/yr, in stark

contrast to the 5391 mm/yr projected by the PM method. The substantial variation in estimated PET values across different vegetation types underscores findings from previous studies (Lu et al., 2005). This study highlights the significance of methodology in computing PET values for hydrological studies and demonstrates considerable spatial variability in estimated PET values among the six evaluated methods. These evidences provide an important scientific foundation for the development of PET methods for water resources management. In addition, some studies examined the contribution of PET

methods to the total uncertainty of PET projections. In Belgium, PET methods show a comparable contribution with Representative Concentration Pathways (RCPs) and Global Climate Models (GCMs) to the total PET uncertainty (Hosseinzadehtalaei et al., 2017). The Hamon method gave higher PET than the PM and PT methods in the North American (McAfee, 2013). In China, the PM method also projected a higher PET than the Hargreaves method (Lemaitre-Basset et al., 2022; Wang et al., 2015)).

**4.1 The key to determine irrigation water requirement (IWR)**

IWR is the water required for irrigation to meet potential ET, percolation, and other water demands, which are not met by the soil water (shown in Fig.4). It is crucial to emphasize that the accurate estimation of IWR is contingent upon local conditions and the availability of data. As of now, the most widely employed approach for calculating IWR is rooted in the water balance method. The pivotal factors for determining IWR encompass a deep understanding of the cultivated crop, an

assessment of soil properties, and a precise knowledge of the crop-specific PET at the given site.

Utilizing the water balance method, the estimation of IWR can be achieved through historical observations or numerical models. However, a significant challenge arises due to the scarcity of comprehensive, long-term databases suitable for this purpose. Additionally, these databases often lack the versatility for extrapolating their findings to different sites, thus limiting their applicability in IWR estimation. Consequently, to circumvent these constraints, numerical models have been

developed and are actively deployed for the precise estimation of IWR across diverse crop and production conditions. Numerical models generally considere plant physiological processes, such as the semi-empirical model (the FAOPM method), process-based model (crop model, hydrological model, and land surface model), remote sensing, and machine learning.

The FAOPM method assumes that there is no occurrence of local advection at the surface. Therefore, meteorological

variables at the reference level z can be considered similar to those in the broader surrounding area (Pereira et al., 1999). Generally, the crop water requirement (referred to as ETc and is equal to PET), is typically estimated by multiplying ET0 by a crop factor (Kc), representing the water needs of a specific crop. The IWR may be calculated as the difference between ET0*Kc and ETa. ETa is usually unknown for most land areas. It is also obviously a challenge to determine the Kc which



depends on the actual phenological state of the crop and the management practice. We notice secondly that it is implicitly
assumed that the meteorological conditions above the reference surface and the specific crop of interest are assumed the
same. These three difficulties (advection, crop factor, and surface feedback to near surface atmospheric conditions) create
lots of other uncertainties in applying the concept of ET0 to practical situations on a global scale. Furthermore, de Bruin et al.
(2016) noted that the FAOPM method excludes local advective heat flux which is the heat supply from neighboring fields
that may suffer from water stress and which may have different crops or surface conditions.

In this study, instead of assuming the same meteorological conditions about the actual surface where the fluxes are measured
as that of a reference grass surface, we derive PET with a validated physical process model (STEMMUS-SCOPE) by
assuming that the said surface is not subject to water stress, while other surface and near surface meteorological conditions
remain unchanged. According to the mass, energy, and momentum balance functions, STEMMUS-SCOPE simulated
evapotranspiration (ETas), soil moisture, and root distribution, and then the IWR can be calculated through eq. (11) to (13).

All in all, the precise prediction of IWR entails a holistic approach, taking into account the intricate interplay of various
factors. This involves the location, soil property, climate, effective rain, cultivation practice, crops and phenology, etc.
(Solangi et al., 2022). When there is no in-situ measurement of ETa in most conditions, the physically consistent model (e.g.
STEMMUS-SCOPE) stands as an exemplary numerical tool for ascertaining IWR in the realms of agricultural and natural
water management.

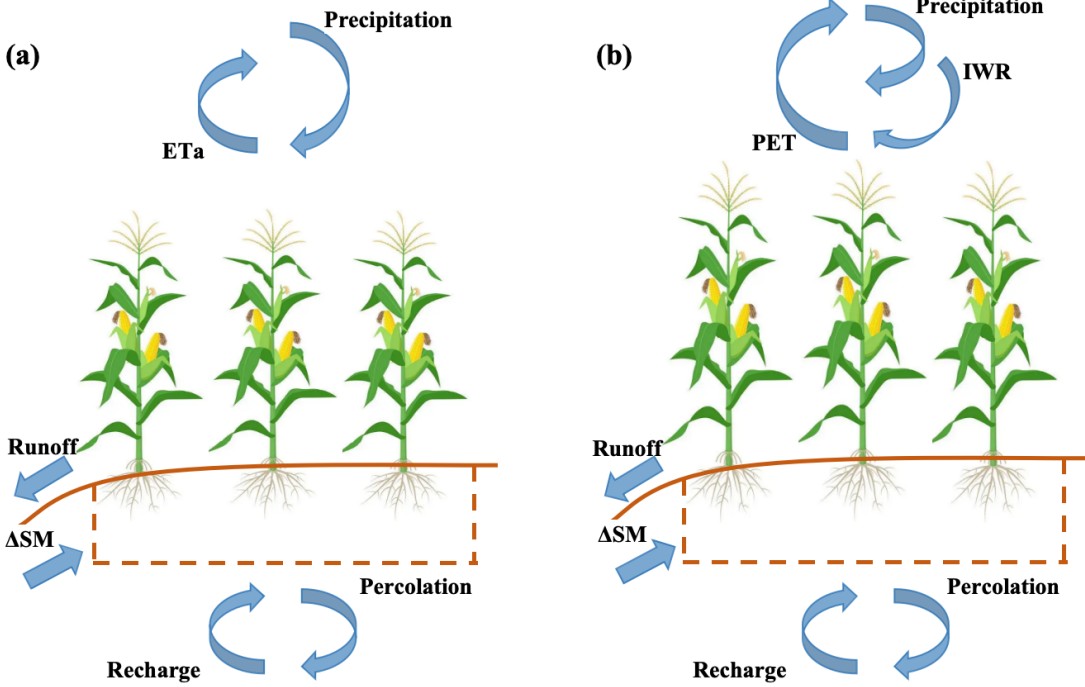


**Figure 4 Diagrams of the soil-water balance of a crop root zone under the (a) rain-fed and (b) irrigation scenarios.**



## 5 Conclusion

Based on a physically consistent model, we evaluated six potential/reference evapotranspiration (PET) methods. These methods were carefully examined for their effectiveness and correlation with the STEMMUS-SCOPE estimated potential
evapotranspiration. Our findings suggest that the PT methods outperform the MAK, HRG, FAOPM, PM, and LSA_SAF methods worldwide, owing to their ability to provide robust estimates while utilizing less data. The PT method, specifically, offers reliable and rational estimates of annual PET.

Of the six methods, we strongly recommend the PT method when available energy (net radiation minus ground heat flux) is accessible. The LSA_SAF method should be chosen when only net radiation is available. Alternatively, if only global
radiation information is provided, the MAK and HRG methods could serve as suitable alternatives. While the FAOPM method can be utilized when wind speed and air humidity data are available, it falls short of being a preferred choice. To summarize, the equations of the PT and LSA_SAF methods demonstrate remarkable suitability for estimating PET on a global scale, while the PM model exhibits significant errors in PET prediction. In essence, the FAOPM method proves inadequate for evaluating the reliability of diverse PET models. Therefor, there are different selection strategies for different
vegetation types. In addition, the actual evapotranspiration (ETa) and the Irrigation water requirement (IWR) can be calculated by STEMMUS-SCOPE and analyzed at different sites.

Through this study, we introduced a practical approach for calculating irrigation water requirements. Moreover, we affirmed the dependability of the PT model, providing pivotal insights for refining existing hydrological models and guidance for agricultural water management.

## 325 Data availability

The PLUMBER2 dataset is available at http://doi.org/10.25914/5fdb0902607e1 (Ukkola et al., 2021). The simulations of STEMMUS-SCOPE model can be acquired from https://zenodo.org/records/11057907. It contains half-hourly energy and carbon fluxes, soil moisture, and soil temperature of 170 sites. These data are stored in NetCDF format with one file per site.

## Code availability

The code of the STEMMUS-SCOPE model can be acquired from https://github.com/EcoExtreML/STEMMUS_SCOPE.

## Competing interests

At least one of the (co-)authors is a member of the editorial board of Hydrology and Earth System Sciences.

## Acknowledgements

This study was funded by the National Natural Science Foundation of China (grant no. 42105119, and 41971033), Dutch
Research Council (NWO) KIC project WUNDER (grant no. KICH1. LWV02.20.004), Netherlands eScience Center project



EcoExtreML (grant no. 27020G07, project ref. no. ASDI.2020.026), and the Netherlands Organization for Scientific Research under Project ALW-GO/14-29.

**Author contributions**

YW, HB, and ZS presented the algorithm. YW, HB and ZS designed the experiment. YW conducted the experiments. YZ, ZS, DY, ET, QH, HB, and ZS revised the manuscript. All authors reviewed and provided comments on the work.

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
