# Peer review of "On the Estimation of Global Plant Water Requirement"

_EGUsphere, 2024_

## Author Comment (AC2)

**Response letter to comments (egusphere-2024-1321)**

**The following is a point-to-point response to reviewer #1's comments.**

We appreciate these comments/suggestions. They would help to significantly improve the quality of this study. Specific revisions and responses to each comment are provided in detail below. Please note that the comments from the reviewer are in *italics* followed by our responses in **regular** text.

**Response to the comments of Reviewer #1:**

**Comment 1**

*The manuscript presents an analysis of six different Potential Evapotranspiration (PET) methods using data from 170 flux sites, with an additional aim to link these analyses to irrigation water requirements (IWR). While the paper extends the dataset size compared to Wouter et al.'s 2019 study, there are significant issues that need addressing before this work can be considered for publication in the HESS journal.*

**Response:** Thank you for your constructive comments. In addition to including more sites in this study, we used a model with physically consistent processes to evaluate common PET methods. We hope the revised manuscript, with a more in-depth analysis about the Irrigation Water Requirement (IWR), will address your concerns.

**Comment 2**

*Clarity of research contribution: The manuscript claims to enhance the understanding and quantification of different PET and IWR estimation methods based on data availability and physical considerations (L23). However, the contribution in terms of IWR is not clearly demonstrated in the results or discussed in the conclusion. It would strengthen the paper to explicitly quantify and discuss the differences and implications of these estimation methods on IWR.*

**Response:** Thank you for your suggestion. Indeed, we overlooked calculating PET after found significant differences within the PET values obtained from these methods. Additionally, because it is very difficult to estimate ETa using these methods, we did not compare the IWR calculated by different PET methods. In the revised version, we will calculate the IWR with the estimated PET by different methods and the observed ETa at each site, and a more detailed analysis of IWR will be presented.

**Comment 3**

*Structure and cohesion: The paper is structured around two main objectives: validating various PET methods and proposing a model for IWR calculation. However, the connection between these objectives is not well articulated, leaving the reader unclear on how they interrelate and support a cohesive thesis. The introduction fails to link these sections logically.*

**Response:** Thanks for your comment. The core of calculating IWR is accurately estimating PET and ETa. This study uses the STEMMUS-SCOPE model, which incorporates physically consistent processes, to calculate ETa and PET, and subsequently IWR. In the revised version, we will reconstruct this section to address this issue.

**Comment 4**

*The conclusion briefly mentions IWR without integrating it into the key findings.*

**Response:** Thank you for pointing out this issue. In the revised version, we will expand on the IWR in the Results, Discussion, and Conclusion sections. Specifically, we will select representative sites to evaluate the effectiveness of the IWR calculated by the STEMMUS-SCOPE model.

**Comment 5**

*Further, the description and justification of the STEMMUS-SCOPE model used are insufficiently detailed. This model is very important as the PET calculated by this model is used as a reference to assess the 6 PET formulas.*

**Response:** Because of the model's structural complexity, we did not provide a detailed introduction and description in this paper. In the revised version, we will provide a clear technical road map (Fig. 1) and include key formulas in the Appendix. Following previous studies that calculate ETa using PET and a water stress factor (WSF), we calculate PET in this study using ETa/WSF.

[Figure]

Figure 1: Technology road-map to generate the data in this study.

**Comment 6**

*Then I am curious about how this model works, if it works better than 6 PET formulas, and how accurate/uncertain this model is. However, this information is largely ignored in this paper.*

**Response:** This study primarily uses model-estimated PET to evaluate other commonly used PET models. Regarding model accuracy, some results at the site scale have already been published (Wang et al., 2021; Tang et al., 2024). As for the simulation results at the 170 sites, we plan to publish it in another paper (is under reviewing by Scientific Data). This is also why we did not elaborate much on the model operation and validation in this paper. This pater only focuses on the usage of the datasets that has already been published on Zenodo (Wang et al., 2024). Overall, the model well simulated Rn, LE, H, GPP, and SM globally and the seasonal variations of fluxes were also captured. Some key figures (Fig. 2 and 3) will be presented in the revised version and are as follows.

[Figure]

Figure 2. (a) Global distribution of PLUMBER2 sites; (b) Performance (Kling–Gupta efficiency, KGE) of STEMMUS-SCOPE (box plots) for the validation set of observations; (c) Performance (KGE) of STEMMUS-SCOPE and GSSM 1km (box plots) for the validation set of observations. The box plots show (from top to bottom) the maximum, 75th percentile, median, 25th percentile, and minimum. The whiskers extend to the most extreme data points not considered outliers, and the outliers are plotted individually using the '○'marker symbol.

[Figure]

Figure 3. Time series of modeled and observed daily LE and H for 170 sites. (The solid line represents the mean daily values of 170 sites, and the shaded area indicates the standard deviation of these 170 sites)

**Comment 7**

*Terminology and definitions: What is the meaning and definition of plant water requirement? It appears once in the title and twice in the abstract, and then disappears in the rest of the text. Is plant water requirement equal to PET, ET0, or IWR? A precise definition of each term at the outset, and consistent usage throughout the paper, are necessary to ensure clarity and professionalism.*

**Response:** Thank you for your comment. In this study, plant water requirements means the IWR. Incorporating Reviewer #2's comments, we plan to use "plant water deficit" instead of "irrigation water requirement". And a clear definition of plant water deficit will be provided in the revised version.

**Comment 8**

*Introduction: The introduction is not well-organized with convincing logical lines, which leads to your research questions. In the whole paragraph from L44 to Line L52, there is no one reference to support your description, which is not a professional way*

*of academic writing. For example, in the sentence "Across various research endeavors, semi-empirical methods and process-based models have demonstrated noteworthy prolificacy in ET0 estimation by leveraging the limited climatic variables as inputs.", references are needed to demonstrate "Across various research endeavors".*

**Response:** Thank you for your suggestion. We acknowledge that the current Introduction section overly emphasizes the computation of PET and neglects its role in calculating IWR. In the revised version, we will reorganize this section to focus on the calculation of IWR first. And we will explain the significance of PET for IWR and highlight the issues associated with current PET methods.

**Comment 9**

*Figures: The figure 1 and 3 suffer from low resolution. The width of the bars in Figure 3 should be the same.*

**Response:** The resolution of figure 1 and 3 have been improved and the font size of figure 3 are enlarged. The width of the bars in Figure 3 are also be keep consistent.

[Figure]

Figure 4. The (a) Coefficient of determination ($R^2$), (b) Mean bias error (MBE, in mm day$^{-1}$), (c) Standard deviation (SD), and (d) Normalized mean error (NME) between calculated daily potential evapotranspiration ($ET0_m$) using different formulas and modeled daily potential evapotranspiration ($ET0_s$) by STEMMUS-SCOPE at 170 flux sites.

[Figure]

Figure 5. The calculated irrigation water requirement (IWR) by STEMMUS-SCOPE for different vegetation types at 170 flux sites.

**Comment 10**

*Summary: The manuscript could have potential but requires significant revisions for research questions, narrative flow, and overall manuscript presentation, particularly in defining and integrating its key components—PET and IWR estimation. I recommend a thorough restructuring to better align this journal's expectations.*

**Response:** Thank you again for your comments on the structure and key components of this manuscript. I hope its readability would be significantly improved with above-mentioned modifications and the revised version could be reconsidered by you.

**References:**

Wang, Y. et al. Integrated modeling of canopy photosynthesis, fluorescence, and the transfer of energy, mass, and momentum in the soil–plant–atmosphere continuum (STEMMUS–SCOPE v1.0.0). Geoscientific Model Development 14, 1379-1407 (2021). https://doi.org/10.5194/gmd-14-1379-2021

Tang, E. et al. Understanding the effects of revegetated shrubs on fluxes of energy, water, and gross primary productivity in a desert steppe ecosystem using the STEMMUS–SCOPE model. Biogeosciences 21, 893-909 (2024). https://doi.org/10.5194/bg-21-893-2024

Wang, Y. et al. STEMMUS-SCOPE for PLUMBER2: A Physically Consistent Dataset Across the Soil-Plant-Atmosphere Continuum (v1.0.3). Zenodo, (2024). https://zenodo.org/records/11323245.

---

## Author Comment (AC3)

**Response letter to comments (egusphere-2024-1321)**

**The following is a point-to-point response to reviewer #2's comments.**

We appreciate these comments/suggestions. They would help to significantly improve the quality of this study. Specific revisions and responses to each comment are provided in detail below. Please note that the comments from the reviewer are in *italics* followed by our responses in **regular** text.

**Response to the comments of Reviewer #2:**

**Comment 1**

*This paper evaluates methods to calculate Potential Evapotranspiration (PET) by comparing them with PET simulated by STEMMUS-SCOPE model at 170 sites. It calculates Irrigation Water Requirement (IWR) as the difference between PET and Actual Evapotranspiration (ETa) simulated by STEMMUS-SCOPE.*

*1.1 It seems that these two objectives are not well connected as the results of PET methods does not add any information to the calculation of IWR.*

*1.2 If the authors aim to use STEMMUS-SCOPE ETa and ET0 to calculate IWR, why needs to validate other 6 methods considering simulated ET0 as the reference?*

*1.3 What is the relevance or significance of calculating IWR for natural land covers (i.e., forest, shrubland)? Why do we need to (know how much to) irrigate these natural vegetation and wetlands?*

**Response:**

1.1: Thank you for your comments on the Introduction section. Reviewer #1 also noted that the comparison of PET methods and the calculation of IWR are not well connected. We will revise these sections for better clarity and connection.

1.2: This study aims to calculate Irrigation Water Requirement (IWR) using a process-based model and to evaluate commonly used PET models. Currently, PET or ETc must be calculated to determine IWR. This leads to significant difficulties and uncertainties in application. Thus, we first evaluated commonly used PET methods.

1.3: We agree that calculating IWR for natural vegetation or wetlands seems less significant. The primary aim of calculating IWR is to evaluate drought severity. Therefore, it is more appropriate to refer to it as Plant Water Deficit (PWR) for the natural sites. We used the term IWR because it is commonly used in agriculture. With

the calculated IWR, we can evaluate the drought condition of each site. In the revised version, we will use PWR instead of IWR.

**Comment 2**

*The paper seems to assume that FAOPM ET0 and PET (of other 5 methods) estimate the same quantity, which is inaccurate by definition. FAOPM is not comparable to other PET methods. It is the only of the 6 methods that calculates PET for a hypothetical reference crop, hence called Reference ET. Even Allen et al. (1998, chapter 1) strongly discouraged using the term PET due to its definition ambiguities, which distinguishes this method from other PET methods. Raza et al. (2022) also showed the difference in definition and purpose of RET (ET0) and PET through a systematic review. Indeed, hydrologists should not use FAOPM RET as a reference to compare with PET by other methods (also mentioned in L58). Then why is FAOPM RET compared with other PET methods here?*

**Response:** As stated in this study, many previous studies use ET0 as a benchmark for evaluating other PET methods. I fully agree with your suggestion to calculate ETc or PET using the crop coefficient (Kc) and then compare it with other PET methods. To ensure clarity, we will consistently use the term PET in the revised version.

**Comment 3**

*This difference in definition might also be the reason for FAOPM to differ greatly from other methods, except when the authors compare them over a vegetation surface with closer characteristics to the FAOPM hypothetical reference crop (e.g., GRA, CRO in Figure S2.1-4). It would make more sense to compare PET methods with FAOPM ETc which is closer to PET definition: the crop evapotranspiration under well-watered and optimal agronomic conditions.*

**Response:** Thank you for your suggestion. We calculated the daily Kc for each site and found significant seasonal variability. Therefore, it is challenging to provide a reasonable Kc for each site, especially for forest sites. In the revised version, we will try to calculate Kc using LAI and climate data. With the calculated Kc, PET can be derived based on the FAO56 method. Thus, the PET calculated by FAO56 PM method can be compared with other common PET methods. For the PM method, we will also adjust the calculation of canopy conductance which currently set as a constant. In the revised version, we will try to calculate it based on LAI.

**Comment 4**

*Figure 2: Why does the distribution of ET0s look remarkably lower than ETas for DBF, ENF, GRA, and MF? Can WSF be greater than 1? By definition, ETa should never be higher than ET0 (Fisher et al., 2010). This makes the method to derive ET0*

*by using STEMMUS-SCOPE simulated ETa and WSF seem not reliable. If so, it should not be used as a reference to evaluate the other PET methods.*

**Response:** Thank you for highlighting this issue. We thoroughly examined the site with these problems and identified two causes: Firstly, the energy constraint was applied on an hourly scale, leading to underestimation at night due to negative available energy. So, we will adjust the energy constraint to apply only during the daytime. Secondly, we found missing values in the Rn or G data while ETa data was available, causing underestimation of ET0 by the STEMMUS-SCOPE. We will solve this issue by omitting these data or using the mean daily value in the next version.

**Comment 5**

*Section 2.5 mentions FAO for the calculation of IWR, but lack references. If the authors referred to the FAO-56 guideline, this method is inaccurate according to the definition in this guideline. Allen et al. (1998) defined IWR = CWR – Peffective, where CWR is ET0 * Kc and Peffective is calculated as detailed in Doorenbos et al. (1977). Therefore, IWR is not the same as ET0 – ETa, which is equal to (1-Kc)*ET0. The equation 9 and 10 seem to be authors' own derivation based on Figure 4. I wonder why the sign of Percolation is negative on the right side of these equation. Shouldn't it have the same sign as Runoff, since they are both 'outflows' from the crop root zone to groundwater storage? Why does Recharge have an upward arrow in Figure 4?*

**Response:**

In this study, the Irrigation Water Requirement (IWR) was calculated as ET0 minus ETa. It should be noted that ETa equals Ks*Kc*ET0, not Kc*ET0, where Kc*ET0 represents ETc in FAO56. Effective precipitation refers to the portion of total precipitation available for plant use, corresponding to ETa. Thus, this study assumes that $P_{effective}$ equals ETa, based on eq. (10): $P_{effective} = P - R + PR - (M1 - M0) = ETa$

"Recharge" refers to the capillary rise from groundwater to the root zone, while "Percolation" denotes water flow from the root zone to groundwater. To clarify, we will revise the terms to:

"PR - Percolation to the groundwater and capillary rise recharge from the groundwater (mm)"

$$P + IWR = ET0 - PR + (M1 - M0) + R \qquad (9)$$

$$P = ETa - PR + (M1 - M0) + R \qquad (10)$$

$$IWR = ET0 - ETa \qquad (11)$$

[Figure]

Figure 1. Diagrams of the soil-water balance of a crop root zone under the (a) rain-fed and (b) irrigation scenarios

**Comment 6**

*Introduction lacks a lot of references. The whole paragraph L44-L56, there's not a single reference to any of the claims.*

**Response:** Thanks, you and Reviewer #1, for pointing this issue. We will reconstruct the Introduction section and add relevant references.

**Minor comments**

**Comment 7**

*L65: FAOPM does not use alfalfa as reference. Maybe you mean ASCE method?*

**Response:** We will delete the "alfalfa" in this sentence.

**Comment 8**

*L66: references?*

**Response:** A reference will be added.

Allen, R.G., Pereira, L.S., Raes, D. and Smith, M., 1998. Crop evapotranspiration-Guidelines for computing crop water requirements-FAO Irrigation and drainage paper 56. Fao, Rome, 300(9), p.D05109.

**Comment 9**

*L71-72: ETas and ET0s. I suggest explaining that 's' means simulated*

**Response:** Thanks. 's' here means the STEMMUS-SCOPE.

**Comment 10**

*L96 & Table 1: suggest to make it clearer which methods are considered radiation-based, resistance-based, global radiation & temperature-based*

**Response:** Thanks for your suggestion. Radiation-based include the PT and LSA_SAF method; resistance-based include the PM and FAOPM method; global radiation & temperature-based include the MAK and HRG method.

**Comment 11**

*Section 2.3.4: How was ga calculated?*

**Response:** The calculation of ga is as follows:

$$g_a = \frac{(K)^2 u_Z}{ln\left[\frac{z_m - d}{z_{om}}\right] ln\left[\frac{z_h - d}{z_{Oh}}\right]}$$

where $g_a$ is the aerodynamic conductance (m/s); $z_m$ is the height of the wind speed measurement (m); $z_h$ is the height of the humidity measurement (m); $d$ is the zero plane displacement height (m); $z_{om}$ is the roughness length governing momentum transfer (m); $z_{Oh}$ is the roughness length governing transfer of heat and vapour (m); $K$ is the von Karman's constant (0.41); $u_Z$ is the wind speed at height $Z$ (m/s).

**Comment 12**

*L123: even if with wet surface, the stomata cannot be as open as a water surface. Therefore, gs cannot be infinity.*

**Response:** I fully agree with you and we used the a constant (70 s m$^{-1}$) in this study. In the revised version, we will try to calculate canopy conductance based on LAI.

**Comment 13**

*L145: references?*

**Response:** We added the reference and a detailed calculation of IWR. Here we used ETa as the effective rainfall. Please find the response to Comment 5.

**Comment 14**

*Figure 3: some bars looks exceeding the maximum value of y-axis. It's not easy to tell if these are much higher than ymax.*

**Response:** The maximum value of y-axis are adjusted.

[Figure]

Figure 3 The calculated irrigation water requirement (IWR) by STEMMUS-SCOPE for different vegetation types at 170 flux sites.